# CBM: An IoT Enabled LiDAR Sensor for In-Field Crop Height and Biomass Measurements

**DOI:** 10.3390/bios12010016

**Published:** 2021-12-29

**Authors:** Bikram Pratap Banerjee, German Spangenberg, Surya Kant

**Affiliations:** 1Agriculture Victoria, Grains Innovation Park, Horsham, VIC 3400, Australia; bikram.banerjee@agriculture.vic.gov.au; 2Agriculture Victoria, AgriBio, Centre for AgriBioscience, Bundoora, VIC 3083, Australia; german.spangenberg@agriculture.vic.gov.au; 3School of Applied Systems Biology, La Trobe University, Bundoora, VIC 3083, Australia

**Keywords:** internet of things, Raspberry Pi, LiDAR, GNSS, high-throughput plant phenotyping, precision agriculture

## Abstract

The phenotypic characterization of crop genotypes is an essential, yet challenging, aspect of crop management and agriculture research. Digital sensing technologies are rapidly advancing plant phenotyping and speeding-up crop breeding outcomes. However, off-the-shelf sensors might not be fully applicable and suitable for agricultural research due to the diversity in crop species and specific needs during plant breeding selections. Customized sensing systems with specialized sensor hardware and software architecture provide a powerful and low-cost solution. This study designed and developed a fully integrated Raspberry Pi-based LiDAR sensor named CropBioMass (CBM), enabled by internet of things to provide a complete end-to-end pipeline. The CBM is a low-cost sensor, provides high-throughput seamless data collection in field, small data footprint, injection of data onto the remote server, and automated data processing. The phenotypic traits of crop fresh biomass, dry biomass, and plant height that were estimated by CBM data had high correlation with ground truth manual measurements in a wheat field trial. The CBM is readily applicable for high-throughput plant phenotyping, crop monitoring, and management for precision agricultural applications.

## 1. Introduction

Crop biomass and height are fundamental morphological traits to estimate crop growth and selection of genotypes of interest in a breeding program. Crop biomass is associated with plant growth and development, being the basis of vigor and net primary productivity [1,2,3]. The crop biomass is a measure of the total fresh weight (FW) or dry weight (DW) of organic matter per unit area [1,4] and is traditionally measured by destructively harvesting plants and weighing for the FW, and oven drying and then weighing again to get the DW. Plant height is the vertical distance from the ground level to the upper boundary of the primary photosynthetic tissues [5,6] and is conventionally measured in the field using rulers. These manual and destructive data collection methods are inefficient, laborious, expensive, prone to manual error, not repeatable, and only applicable to small scale field trials.

Proximal digital sensing technologies overcome such challenges by offering a practical solution for high-throughput plant phenotyping (HTPP) of crop biomass and height [7,8]. There are several sophisticated and multi-sensor HTPP systems, including Field Scanalyzer [9], Ladybird [10], Phenomobile [11], and Phenomobile Lite [12], that have been developed for field phenotyping applications. Over the past decade, different automatic sensing techniques have evolved to measure crop biomass and height as a quantitative trait under field conditions using imaging devices and ranging sensors [13]. Imaging devices such as structured-light scanners [14], stereo cameras or stereo vision systems [15,16], and structure from motion systems are principally based on photogrammetric triangulation, which often poses heavy computation overload. Different reflectance that is based imaging sensors such as multispectral and hyperspectral systems have been used to estimate the biomass. The fusion of reflectance and structural information has been reported as being useful to estimate the crop biomass [3,17]. Other sensors, such as ultrasonic devices and most of the light detection and ranging (LiDAR) systems, are all based on the time-of-flight (ToF) principle [14,18].

ToF range sensors can directly provide the distance of the target object (i.e., the plant canopy) from the detector to measure crop height. Ultrasonic sensors as ranging devices can provide crop height and biomass measurements that are well correlated with the ground truth measurements, if finely calibrated [19,20]. Such devices detect objects within their operating sound cone and provide an averaged distance, i.e., inherently filtering the variable crop height within a plot [19]. However, selecting a suitable field-of-view (FoV) is necessary to cover sufficient plant canopy area without exceeding the scanning footprint that is outside the canopy cover to avoid detecting non-plant objects [21]. Additionally, the ultrasonic sensors are sensitive to temperature as sound speed changes with temperature [22] and sound waves are susceptible to plant leaf size, angle, and surfaces [23]. Ranging devices that employ pulsed light compared to sound echoes are technologically faster, providing better a sampling frequency, which is essential when distance is being measured from a mobile field vehicle. A relatively new technology, ToF cameras, were developed to capture color and distance as image arrays simultaneously. These ToF cameras have been found to be promising for 3-dimensional (3D) crop height measurements recently [24,25]. However, such cameras are sensitive to direct sunlight and require shading to provide accurate measurements [25]. Conventional LiDAR sensors are relatively effective in direct sunlight and can capture a high-resolution representation of the 3D canopy structure [26,27] and the crop biomass and height can be extracted through post-processing. Although LiDAR is widely accepted and a promising sensor for phenotyping crop biomass and height, it is also costly and LiDAR data can be huge and difficult to process [28].

Internet of Things (IoT)-based systems benefit real-time decision-making with a high level of automation. These systems are becoming more affordable and available for various fields, including agricultural plant phenotyping [25,26,27]. Additionally, the system architecture of IoT systems mostly consists of open-source packages and software which support a wide range of interfacing solutions and options to adapt and extend implementations in different experimental scenarios [28]. Raspberry Pi (Raspberry Pi Foundation, Cambridge, UK), a low-cost system-on-chip computer, has been used in broader IoT applications. The system has been used for collecting high-quality imaging data in different scientific and engineering applications [29,30,31]. In plant phenotyping, such systems were used in applications such as the tracking of leaf movement using time-lapse images [30], investigation of crop disease response in controlled environment conditions [32], evaluation of crop health [33], and in-field crop monitoring [34].

To this end, this research developed a low-cost sensor that was named CropBioMass (CBM) with simplified data acquisition and processing and the seamless extraction of plant traits measurements crop biomass, and height. The CBM was packaged with a solid-state LiDAR sensor for range measurements, an onboard computer with WiFi connectivity to the internet for easy data transport to a cloud platform, and a global navigation satellite system (GNSS) logger to measure the spatial geolocation of the range measurements. This study reports the design implementation details, including the system architecture, 3D design, and the software architecture processing workflow of CBM. Field testing was conducted on a wheat field trial containing multiple genotypes and results showed that the crop fresh- and dry-biomass and the plant height that was estimated by CBM had a high correlation with the manually measured data.

## 2. Materials and Methods

### 2.1. System Architecture

There are three main hardware components of IoT that enabled the CBM sensor; solid-state LiDAR sensor (Leddar Tech Vu8, Leddar Tech, Quebec City, QC, Canada), onboard Raspberry Pi 4 computer, and autopilot controller that was used for an embedded GNSS logger (Navio2, Emlid Ltd., Hong Kong). 

For the crop height measurements, a solid-state LiDAR LeddarTech Vu8 module was used as it specifically measures ranges in a given direction within a required FoV. Regular commercial-grade LiDARs scans in every direction (360°) that produce voluminous data and pose processing challenges. A narrow FoV solid-state LiDAR unit is suitable for scanning only the canopy profile which significantly reduces the data output and processing overload. The selected Leddar Tech Vu8 module sends pulsed light in a horizontal FoV of 48° with the reflected pulses measured into 8 discrete segments each with 6° of horizontal FoV coverage; the vertical FoV of the module is 0.3°. The laser emitter operates over a near-infrared wavelength of 905 nm with IEC 60825-1:2014 Class 1 eye safety rating. It works with a 12 V ± 0.6 DC input power supply with a power consumption of 2 W. LeddarTech Vu8’s carrier board hosts the electrical and communication interface of the module and has two interfacing options, including SPI or USB-CAN-serial (UART/RS-485). The sensor has a programable data refresh rate, measurement accumulation, and sensitivity peak, which help in tuning the sensor to accurately profile different type of targets, including vegetation. 

A Raspberry Pi 4 4 GB Model B single-board computer (SBC) was used as the sensor driver unit. Raspberry Pi is known for its compact size (as small as the size of a credit card) and relatively good processing power- 1.5 GHz quad-core Cortex-A72 (ARM v8) 64-bit System-on-Chip (SoC). It has 4 GB of LPDDR4-3200 Synchronous dynamic random-access memory (SDRAM) storage, 2.4 GHz and 5.0 GHz IEEE 802.11ac wireless (Bluetooth 5.0, BLE), 2 USB 3.0 ports, 2 USB 2.0 ports, 40 pin general-purpose input/output (GPIO) header, and a micro-SD card slot for loading operating system and data storage. The SBC can be powered with a 5 V DC via USB-C connector or GPIO header. The Raspberry Pi also have low power consumption and plethora of available library packages, which is beneficial in rapid prototyping of new sensor systems. For tagging the LiDAR measurements with positioning logs, a Navio2 GNSS receiver (Emlid Ltd., Hong Kong) was used supporting GPS, GLONASS, Beidou, Galileo, and SBAS satellites constellation systems. Additionally, Navio2 is relatively low-cost, more reliable than other commercial-grade positioning sensing systems.

A detailed schematic overview of the CBM components is provided in Figure 1. The LeddarTech Vu8 module was interfaced with the USB port on Raspberry Pi 4 over USB-CAN-serial communication. The GNSS receiver on Navio2 was interfaced over the Raspberry Pi 4 with the 40 pin GPIO port. The power supply to the respective units was provided through a voltage regulator circuitry. The regulator provides a steady and filtered DC output power; a 12 V constant output was supplied to the LeddarTech Vu8 module and step-down 5 V output to power the Raspberry Pi and Navio2 modules. The total current consumption was rated around 800 mA in the regular mode of operation.

### 2.2. System 3D Design

An enclosure for the integrated CBM sensor was designed using the Fusion 360 software (Autodesk Inc., San Rafael, CA, USA). The design was fabricated using a 3D printer (Ultimaker S5, Geldermalsen, The Netherlands) with acrylonitrile butadiene styrene plastic filament (Ultimaker, Geldermalsen, The Netherlands). The system’s complete set-up was made up of the following six parts: top, bottom, front, back, left, and right pieces (Figure 2a). The 3D model was designed and produced for each side separately for quicker 3D printing and to avoid the requirement of an additional support filament. Moreover, with all the bodies laid flat on the build plate (Figure 2b), the cross-sectional adhesion between the layered threads of the filament is higher compared to printing models vertically; this produces stronger printed parts. The list of the primary parameters for the 3D printer is provided in Appendix A. Upon printing, the individual parts were assembled using M3 bolts (Figure 2c). 

### 2.3. Software Architecture

The Linux-based operating system, Raspbian Buster, was installed on the CBM sensor. A combination of C++ programing and the open-source Python 3 (Python Core Team 2015) was utilized as the programming languages to implement all the data acquisition functionalities, onboard processing, and cloud uploading.

Once the CBM sensor is switched ON it automatically connects with a predefined WiFi network that is available in the field, to connect with the master repository, which runs on a remote server over the internet. Inside the server, all the high-level functions of further filtering (denoising), processing, and analyzing the acquired data are scheduled immediately after injecting a new acquisition. These high-level processing nodes were implemented on the remote server because of the availability of more processing power. In the current development stage, the Raspberry Pi logs on to a password-protected hotspot upon the boot-up. In the absence of a nearby WiFi network, the CBM sensor performs in a non-networked mode, i.e., the acquired data is saved onto the device’s memory and uploaded to the cloud server once an internet connection has been found. Once the data are uploaded, the system compresses the local copy of the data in an archive for safe-keeping; older data points are routinely deleted as the system space fills to free up system space.

The LiDAR and GPS units take around 20 s to initialize, load all the required packages, and establish a serial connection with the Raspberry Pi. An indicator LED on the system that is mapped to a physical GPIO pin is used to relay the runtime status of the system to the user. At initialization, the system indicator LED remains in a ‘steady solid’, i.e., HIGH state. Upon initialization, it starts ‘flashing rapidly’, i.e., HIGH-LOW state at a frequency of 20 Hz. At this ‘ready’ state, the data acquisition from the LiDAR and GPS could be triggered using a push-button switch (active-low with internal pull-up resistance). During data acquisition, the system status LED reverts to the ‘steady solid’ state. The sensor provides the ability to pause (inactive state) or resume (active state) multiple times, triggering the status LED to change response accordingly. All the critical settings, such as sampling frequency, signal strength, accumulation rate, and time tag format, are predefined as the default parameters for ease of in-field operation. 

### 2.4. Laboratory Sensor Calibration

The laboratory distance calibration experiment was conducted at the SmartSense iHuB, Agriculture Victoria, Horsham, VIC, Australia. The laboratory trial was designed to calibrate the capabilities of the LiDAR under ideal conditions, mounted on a platform with the CBM sensor pointed towards a hard, flat ground surface (bigger than the minimum FoV of the sensor). The height of the platform carrying the CBM sensor was adjusted at different reference distances from the ground surface (Figure 3). The sensor was triggered for approximately 5 s at nine specified positions ranging from 40 cm to 265 cm in steps of approximately 30 cm, collecting 9 points × 8 detectors = 72 readings. The LiDAR unit was calibrated against these reference measurements. Hereafter, these calibration settings were internally applied on the Raspberry Pi to tune the measurements that were captured from the LiDAR unit to provide precise distance profiles automatically. 

### 2.5. Field Experiment and Data Collection

The developed CBM sensor system was deployed in a wheat field trial at the Plant Breeding Centre, Agriculture Victoria, Horsham, Victoria, Australia. The field trial comprised of 36 wheat genotypes that were planted in individual plots (Figure 4a) with a sowing density of approximately 150 plants per m^2^. The list of wheat genotypes is provided in Appendix A. Each plot was 1 m wide and 5 m long (Figure 4a). Field observation that included both the sensor readings and ground truth manual measurements were performed at 100 and 140 days after sowing (DAS) where the plants were at first-node and anthesis growth stages, respectively. 

The CBM sensor was mounted in a downward nadir facing angle. The field mount was a manual push-type vehicle with a wheelbase or width of 1.25 m, which enabled traversing the sensor along the lengthwise direction of the wheat plots (Figure 4a,b). A sinuous traverse path trajectory was used to cover each row in an individual range, followed by the next range in a reverse direction. This pattern enabled the least movement of the rover or the equivalent time in data acquisition. The LiDAR data scan was recorded continuously throughout the scanning mission at a rate of 60 Hz. The field phenotyping data using the CBM sensor was captured on the same days before the manual plant height and biomass harvest. The manual crop height was measured from the ground level to the highest point of the plant with the average of the four random height measurements per plot. The LiDAR sampling method is arguably more accurate than the manual crop height measurement method, at least for assessing average crop height within each plot of ~750 individual plants (150 per m^2^). Thereafter, each plot was manually harvested separately for the above-ground biomass and weighed soon after harvest to measure the fresh weight (FW) biomass and oven-dried at 70 °C for five days to measure the dry weight (DW) biomass.

### 2.6. Scanning Geometry and Crop Height

The CBM sensor was mounted on the mobile platform at the height of 1.8 m (Figure 5). The LiDAR scans were collected with a horizontal FoV of Φ = 48° in the sensor’s trajectory across-track direction. In this configuration, each of the 8 stacked detector elements provided a horizontal FoV of θ = 6° covering the plots along the widthwise direction, AB (Figure 5b,c). The raw range measurements for each detector (r) were recorded by the CBM in .csv format, which was converted into crop height measurements (*h*) using the trigonometric calculations in Equations (1)–(3):(1)h= D − r·cos(β)
(2)β =(θ/2)+ α
(3)θ = Φ/n
(4)w = r·cosβ{tanα−tan(α+θ)}
where, D is the mounting height of the sensor, r is the range measurements that were collected for a detector, α and β are the orientation angles for a detector element from the vertical axis (SH) to the detector’s FoV edge and center, respectively, θ is the width of each detector element, Φ is the total FoV, *n* is the number of detector elements, and w is the width that was traced by a detector with a FoV of θ at an angle of β from the vertical axis on the height (*h*).

In programming terminologies, D, θ, and Φ are constant parameters, whereas, the variables r, α, β, *h*, and w are packaged within equivalent array vectors **R**, **A**, **B**, **H**, and **W** respectively. Therefore, for the case of height (*h*), the array vector **H** = [*h*_1_, *h*_2_,… *h*_*n*_].

### 2.7. Signal Processing for Plot Segemnation, Extraction of Plot Equivalent Mean Crop Height, and BioVolume

The LiDAR scan was collected continuously over the plots along the transects (Figure 4a). The LiDAR scanner picks up spurious or noisy disturbances during the data acquisition due to the undulating ground surface. A Savitzky-Golay filter removed most noisy disturbances due to ground undulation. However, certain disturbances due to the ground undulation remained in the form of false peaks near the edges of the plots (Figure 6a). To further remove these, firstly, the ground surface was classified and masked-off using a vertical threshold (V_th_ < 5 cm) criterion. After that, a horizontal (H_th_) threshold criterion was used to remove such falsely recorded peaks or ‘continuous peak segments’ under the lengthwise scan size of 50 samples (H_th_ < 50 samples). As the scan size or scan profile for the 5 m long plots was significantly greater than the size of false peaks, the H_th_ forms a simple and effective basis to eliminate any slight undulations in the signal profile. Hereafter, the plot profiles were automatically segmented along the direction of the scan (Figure 6b). 

Although the push-type mobile platform or vehicle was traversed at a near-constant speed (1.4 m/s), maintaining a uniform speed throughout the scan duration was not possible (as it is impractical) in field conditions. This leads to variable lengths of the scans for different plots. As data are collected at a constant rate from the LiDAR (60 Hz in this case), the plots over which the vehicle speed was slightly slower captured a greater number of samples, and vice-versa. The number of samples for all the plots ranged between 210 and 400 (Figure 6c). To adjust for this discrepancy, the lengthwise acquisition of the LiDAR samples was resampled at a constant rate of *m* = 100 samples per plot of 5 m in length (Figure 6d). Therefore, the array vector H is acquired m times, or the resampling frequency for each plot and *n* = 8 samples along the plot width direction, forming a *m* × *n* matrix, represented by **H_plot_** in Equation (5).
(5)$$H_{plot}={\left[\begin{array}{ccc} h_{11} & \ldots & h_{n1}\\ \vdots & \ddots & \vdots \\ h_{m1} & \ldots & h_{mn} \end{array}\right]}_{m\times n}\;$$

As the detector elements are stacked at different angles (α, β) from the nadir, the widthwise footprint (w) increases progressively away from the nadir and reduces as the height (*h*) of the crop increases (Equation (4)). For an optimum medium-sized plot, the FoV of the sensor covers the entire width of the plot (Figure 7a,d). However, a short-sized plot is farther from the sensor, so its FoV covers beyond the width of the plot (Figure 7b,e). These segments on the edge are required to be ignored. Inversely, tall-sized plots are closer to the sensor so their FoV covers a limited width of the plot (Figure 7b,e). Therefore, extrapolation is required to account for the missed plants on the edge of the plots. A spatial resampling in the widthwise direction was applied to adjust for this mismatch. The dimension of the **H_plot_** matrix remains unchanged in this step but the internal values are revised.

The **H_plot_** formulates the mathematical matrix formulation of the collected height profiles for each plot. The segmented **H_plot_** matrices were geolocated using the corresponding tags that were collected through the onboard GNSS receiver. The sensor system integrated the Navio2 GNSS receiver to configure the internal clock in sync with the GNSS time. Since the internal GNSS update rate was 5 Hz and did not have DGPS capabilities, there was not enough accuracy for geolocating the height profile, i.e., the entire array vector **H**, that was acquired over the plots. Instead, the GNSS timestamps over each plot provided the geolocation that was needed to register the segmented plot matrix **H_plot_**. 

The geolocated **H_plot_** for the individual plots enabled further geospatial analysis to summarize the average height and volume of the crop, termed as BioVolume (BV). Therefore, the average height (*h*_avg_) of each plot is calculated according to Equation (6).
(6)$$h_{\mathrm{avg}}=\frac{1}{m\cdot n}\overset{s=m}{\underset{s=1}{\sum }}\overset{d=n}{\underset{d=1}{\sum }}h_{m,n}$$
where, *h_m_*_,*n*_ is a specific element of the **H_plot_** matrix at *m*th row and *n*th column, d corresponds to the detector element, *n* is the total number of detectors (*n* = 8), s represents the sample, and *m* represents the total number of samples per plot after resampling (*m* = 100).

The BioVolume (BV) for each plot can be calculated according to Equation (7).
(7)$$\mathrm{BV}\;=\overset{y=Y}{\underset{y=1}{\sum }}\overset{x=X}{\underset{x=1}{\sum }}h_{m,n}\cdot w_{m,n}\cdot t$$
where, *h_m_*_,*n*_ is a specific element of the **H_plot_** matrix at *m*th row and *n*th column; *x* and *y* specifies the width (across-track) and lengthwise (along-track) directions of the plots, respectively; X and Y represent to the dimensions of the plot, i.e., the width and length, respectively; w*_m_*_,*n*_ is the unit width that is traced by an individual detector at height *h_m_*_,*n*_; and t = 5/100 m is the length that is traced by the detector according to the resampling. 

Data that are pushed in through the IoT-based CBM sensor were processed automatically in real-time for plot-level trait extractions. All of the analytical procedures were implemented on a server hosting the IoT-based data injection. The analytical protocols were developed in Python 3.7.8 (Python Software Foundation, Python Language Reference) using source packages including os, fnmatch, matplotlib, numpy, skimage, and opencv2.

### 2.8. Evaluation of the CBM Sensor in Measuring Crop Height, Fresh Weight, and Dry Weight

The CBM sensor-derived results were compared with manual field measurements of the crop height, FW, and DW within each plot. Simple linear regression was used to establish and test on the empirical relationships. Parameters such as coefficient of determination (*R*^2^) coefficients, with their corresponding root mean square error (*RMSE*) and the mean absolute error (*MAE*) were used in the evaluation for best fit models. The *R*^2^, *RMSE*, and *MAE* for each model were obtained according to Equations (8)–(10), respectively.
(8)$$R^{2}=\frac{\sum_{i}\left({\hat{y}}_{i}-\bar{y}\right)}{\sum_{i}\left(y_{i}-\bar{y}\right)}$$
(9)$$RMSE=\sqrt{\frac{\sum_{i}{\left({\hat{y}}_{i}-y_{i}\right)}^{2}}{N}}$$
(10)$$MAE=\frac{\sum_{i}\left|{\hat{y}}_{i}-y_{i}\right|}{N}$$
where $y_{i}$ is the actual value for *i*th observation, $\bar{y}$ is the mean of the actual value of observations, ${\hat{y}}_{i}$ is the predicted value for *i*th observation, and *N* is the number of observations.

## 3. Results

### 3.1. Laboratory Sensor Calibration Results

For sensor calibration, a total number of 72 observations were collected at a variable step distance of approximately 30 cm between 40 cm and 265 cm. The regression line that was plotted between the reference and measured distances in the laboratory is shown in Figure 8. The correlation was very high between each of the sensor’s detector output ranges and the target distance. The fitted models for all detectors explained a 98.5% of the variability of the response. The *R*^2^ for all models averaged 0.98 with an *RMSE* of 3.97 cm, which is very practical for the end-use case of the sensor in phenotyping plant canopies with fragile structures. The average error in absolute value was 3.54 cm.

### 3.2. Estimation of Crop Height, FW, and DW in Field Trial

The crop height, FW, and DW regression models were validated with manual data that were collected at 100 and 140 DAS. The crop height of the wheat genotypes in the experiment ranged from 42 to 92 cm on 100 DAS and 64 to 102 cm on 140 DAS with a normal distribution. The mean crop heights were 80 and 84.6 cm on 100 DAS and 140 DAS, respectively. A correlation-based assessment was performed to evaluate the CBM sensor-derived height readings (h_avg_) with respect to the manual plot height measurements (Figure 9a). The assessment achieved a strong linear relationship between the sensor height readings and the manual crop height with an *R*^2^ of 0.79, *RMSE* of 6.09 cm, *MAE* of 5.03 cm, and a ratio of *RMSE* and *MAE* of 1.21. Unlike the highest points that were measured during the ground-based surveys, the sensor height readings represent the complete relief of the crop surface; therefore, it was found to be about 17.5 cm lower than the average canopy height of the manual measurements.

The crop volume measurement of wheat was earlier reported to correlate with FW and DW [17]. Therefore, linear regression modeling was applied to estimate the FW and DW from BV, producing an *R*^2^ of 0.70 and 0.84, *RMSE* of 0.49 Kg and 0.14 Kg, and *MAE* of 0.38 Kg and 0.11 Kg, respectively (Figure 9b,c). The ratio of *RMSE* and *MAE* was 1.29 and 1.27 for FW and DW, respectively.

A frequency-domain analysis of the residual absolute error showed that the 50% of the observations had an absolute error ≤4.28 cm for crop height, ≤0.28 Kg for FW, and ≤0.09 Kg for DW, when compared with the regression line that was established by the fitted linear model (Figure 9d–f). The accuracy in measuring BV in field conditions influenced the final estimated FW and DW biomass. For example, when estimating canopy BV with a LiDAR sensor, a few centimeters may have some relative effect on the estimation of FW and DW. Nevertheless, the stacked LiDAR unit that was used here had benefit for better capturing the canopy’s spatial profile, which is otherwise challenging with one ultrasonic sensor. The biomass gain between the two time points is also visible in the plots (Figure 9b,c). The fitted linear correlation line follows through the sample distribution at each time point, demonstrating the applicability of the methods in progressively measuring the biomass estimates. 

### 3.3. Phenotypic Screening of Wheat Genotypes

The key objective in plant breeding research is to screen genotypes to select better-performing lines with higher growth and yield potential. Non-destructively collected crop height and biomass estimates are crucial contributors in effectively mapping the growth profile. The crop height, FW, and DW that were estimated across the two time points showed growth trends for the wheat genotypes (Figure 10). The growth height was maximum for Aus27919 between the two time points i.e., 110 and 140 DAS. The genotypes Aus428 and Aus79 achieved the maximum height of 102 cm amongst all other genotypes on second time point, 140 DAS (Figure 10a). In terms of biomass, the genotype Aus7992 had the highest FW and DW of 4.5 Kg and 2.1 Kg, respectively, amongst all other genotypes on 140 DAS. The genotype Cara was shortest and produced minimum FW and DW on 140 DAS (Figure 10b,c).

## 4. Discussion

### 4.1. Developmental Aspects of CBM Sensor

This study used open-source hardware and software platforms to develop CBM sensor for capturing key phenotypes, i.e., crop FW, DW, and height. Specifically, the system development incorporated the integration of a Raspberry Pi 4 board as the central processing unit, LeddarTech Vu8 module as the LiDAR unit, and Navio2 as the GNSS unit. The system design that was used in this study offers modularity of components. There are several alternatives to the onboard processing option, including different models or versions of Raspberry Pi and Arduino boards. Similarly, the sensing modules such as LiDAR and GNSS could be switched with other industry equivalents. Research in this space is beneficial to develop a specialized sensing system for the desired phenotypic traits in different crops.

This provides a developmental platform to embed new analytical packages to simplify the process of high-throughput phenotyping which is otherwise challenging with closed-off-the-shelf available systems. Several studies have demonstrated the integration of sensing systems for phenotyping research. For example, the Raspberry Pi board with a camera to collect images for crop monitoring in sentry mode in field [34], to measure leaf area in field [35], to develop a plant imaging chamber [33] a seed germination phenotyping system [36], and the Microsoft’s FarmBeats system for soil sensing [37]. 

The CBM sensor that was developed in this study can operate for up to three hours on a portable 2500 mAh battery which can be swapped to extend the in-field operational time. In terms of data management, an onboard data storage system with cloud uploading was designed. An approximately 32 GB internal memory card was used in this study, of which 6 GB was invested in system files, packages, and the developed codes, and about 26 GB remains dedicated for data storage. The estimated data size is about 400 Kbytes/minute; therefore, the storage would last for up to approximately 50 days in a continuous mode of operation without cloud uploading; but when the data is regularly uploaded to the cloud, the storage space is automatically cleaned-up even with the local archiving, which provides unlimited practical storage. Furthermore, with all the processing operations that are performed on the server nodes, the local computation is readily avoided, preventing overheating or power drain. The current version of the CBM sensor depends on a WiFi network for internet connectivity. In a future version, a dedicated internet bridge hardware, i.e., subscriber identity module (SIM) card slot could be implemented to ease field use.

### 4.2. Effect of Sensing Geometry and Crop Canopy Effect

The LiDAR sensor needs to detect a dense number of leaves in a vertical plane to estimate the crop height correctly. A smaller number of leaves towards the top of the canopy is unlikely to produce a substantial LiDAR return pulse due to less dimension. Instead, a subsequent return pulse could be produced further down from inside the canopy, where the density of the leaves is higher. In such situations, a shorter height is measured by the LiDAR than the one that is estimated by the regression model, and thus, the measurement falls under the regression line (Figure 9a).

It is also possible that crop canopies contain gaps in the outer layer of leaves. When the sensor’s detector is aligned over these gaps, the returns are likely to be produced by the surrounding canopy leaves. Therefore, the measured distance return remains above the regression line. The scanning geometry of the sensor’s individual detector with respect to the target and its FoV determines the nature of the LiDAR return pulse. As the detector’s FoV varies with the distance to the object, it is challenging to detect gaps in the canopy at large distances. A smaller FoV for individual detectors could be selected to obtain a better representation of the canopy structure. However, such systems are more expensive, and require more storage and computational power to process the large data, limiting its seamless adoption in phenotyping. Therefore, selection of a suitable LiDAR unit for phenotypic application needs to consider such practical trade-offs. While the CBM sensor produced good returns against a flat surface, the increase of *RMSE* in wheat crop suggests that variability in the response is higher. In general, the height that was estimated from the CBM sensor was slightly lower than with manually measured heights with rulers in the field wheat experiment, although this relative difference was consistent across the genotypes. 

The ground-based laser scanners are prone to variation in FoV coverage at the canopy level due to changes in the crop height. Increasing the mounting height assures the sensor’s FoV to cover the entire plot, but it might reduce the scanner’s ability to map the subtle variations within the plot due to a lower resolution. The mounting height of the sensor needs to be adjusted with practical settings such as the average crop height and width of the plots. In this study, the CBM sensor was mounted at the height of 1.8 m and used analytics to trim or extrapolate its FoV to match with the plot width at the canopy level.

### 4.3. Estimation of Crop Biomass and Height

LiDAR sensors that are mounted on ground-based platforms provide consistently better plant height estimates when compared to ultrasonic sensors [21]. For instance, the ultrasonic sensing has been used to estimate the height of cotton [20,38], alfalfa [39], wild blueberry [40,41], legume-grass [23], Bermuda grass [39], barley [42], and wheat [39], with an *R*^2^ between 0.44 to 0.90 and *RMSE* between 0.022 m to 0.072 m. Whereas, LiDAR has been employed for crops such as cotton [43], blueberry [44], and wheat [9,10,11,12], achieving an *R*^2^ between 0.86 and 0.99, and *RMSE* between 0.017 m and 0.089 m. Similarly, LiDAR sensors have been employed for estimating the biomass in several crops, such as cotton [45], arctic shrubs [46], and wheat [12]. Overall, the LiDAR-based systems provided better accuracy in phenotyping the crop genotypes compared to the ultrasonic systems. The higher sampling rate, multiple numbers of stacked detectors, and focused FoV are a few advantages of LiDAR over ultrasonic systems, invariably benefiting the precision of LiDAR-based systems. The integrated CBM sensor that was developed in this study produced significantly less voluminous data, which is both easy for cloud uploading and processing. Therefore, the use of a ground vehicle-mounted IoT-based LiDAR with a cloud processing method could be more appropriate for crop breeding programs when high fidelity, quicker, and seamless processing is required.

### 4.4. Benefits of CBM in Plant Phenomics and Precision Agriculture

Digital sensors for plant phenomics research continue to advance and boost crop genetic improvements for resource optimization, gains in yield, and overall quality. The significant outcome of this study is the ability to non-destructively estimate the plant biomass and height using an integrated ground-based sensor with an end-end pipeline in data acquisition through to IoT-based cloud uploading and processing. The development of new sensors provides opportunities for researchers to assess precise phenotypes or quality traits with a high temporal resolution, which can be used in selecting genotypes with desired performance. Moreover, high temporal resolution data provides the opportunity to study dynamic crop responses to the environment to evaluate genotype performance.

In the context of precision agriculture, crop biomass and height are valuable traits for making informed management decisions. Biomass measurements require physical destructive harvesting whereas proximal sensing systems are able to estimate the same without damaging the in-season crop. The CBM sensor that was developed in this study can be readily mounted on a tractor or boom-spray to collect field measurements. The adopted agronomic design of the small-scale field experiment enables direct transferability of the established biomass and height estimation models to a conventionally managed larger-scale farmer’s field. Furthermore, the presented method of modeling biomass in wheat could be suitably extended for non-destructive in-season estimation of biomass in other grain and forage crops.

## 5. Conclusions

In-field high-throughput plant phenotyping is essential to characterize crop populations accurately and efficiently. This study reports the development of CBM, a low-cost integrated sensor system with LiDAR to measure crop biomass and height in the field. Additionally, CBM can operate as an IoT device to enable remote and direct uploading of the captured data on a cloud platform for further processing and analysis. The CBM sensor was deployed on a wheat field trial to evaluate the accuracy and operational efficacy. The study achieved significant capabilities by developing a unique integrated sensor system that was assembled from widely available hardware resources and using open-source software packages. The system is promising for crop monitoring and estimating growth rate for the evaluation of genotypes. Overall, the CBM sensor produced a reliable performance in collecting phenotypic data in the field. The sensor provides researchers access to an affordable high-throughput digital technology with a complete end-end pipeline.

## Figures and Tables

**Figure 1 biosensors-12-00016-f001:**
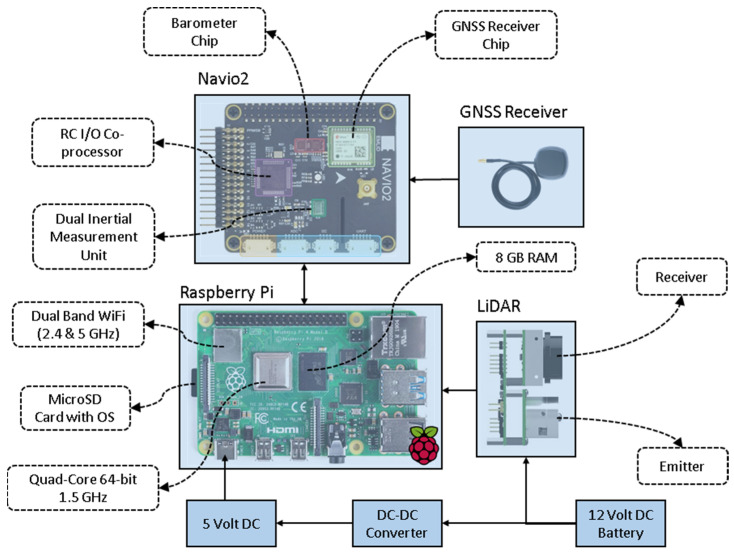
Schematic overview of the components that were used in the CropBioMass (CBM) sensor. The integrated systems consist of a solid-state LiDAR sensor, onboard Raspberry Pi 4 computer, and an autopilot controller that was used for the embedded GNSS logger.

**Figure 2 biosensors-12-00016-f002:**
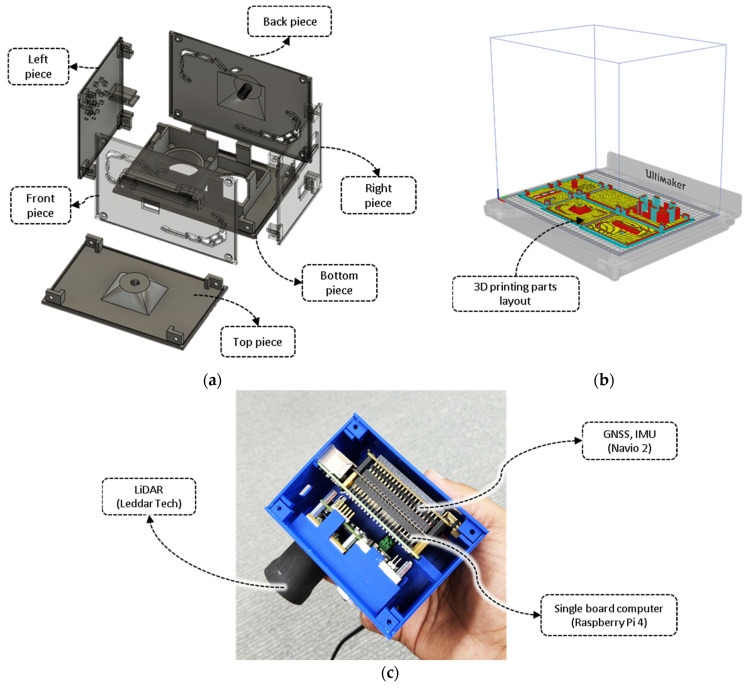
The CropBioMass (CBM) sensor design and development. (**a**) 3D model parts for the sensor case, (**b**) printing of the modeled parts using a 3D printer, and (**c**) encasing of the hardware units inside the printed 3D model case.

**Figure 3 biosensors-12-00016-f003:**
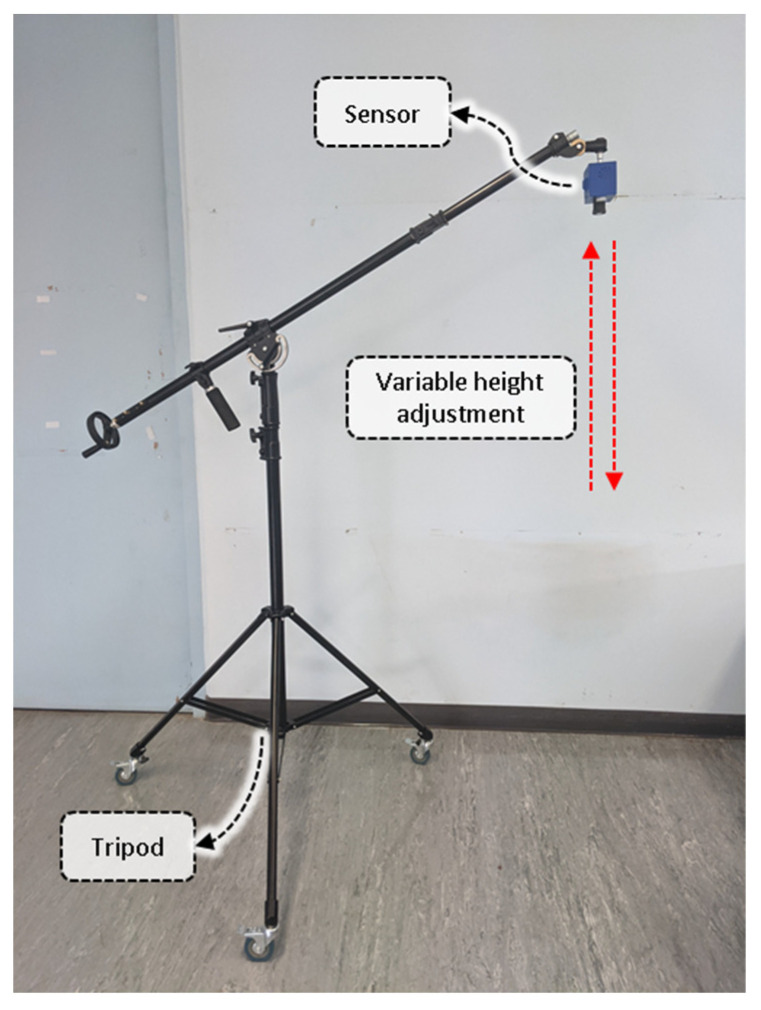
The laboratory distance measurement experimentation. The CropBioMass (CBM) sensor was positioned at different reference distances. The LiDAR measurement response was empirically calibrated.

**Figure 4 biosensors-12-00016-f004:**
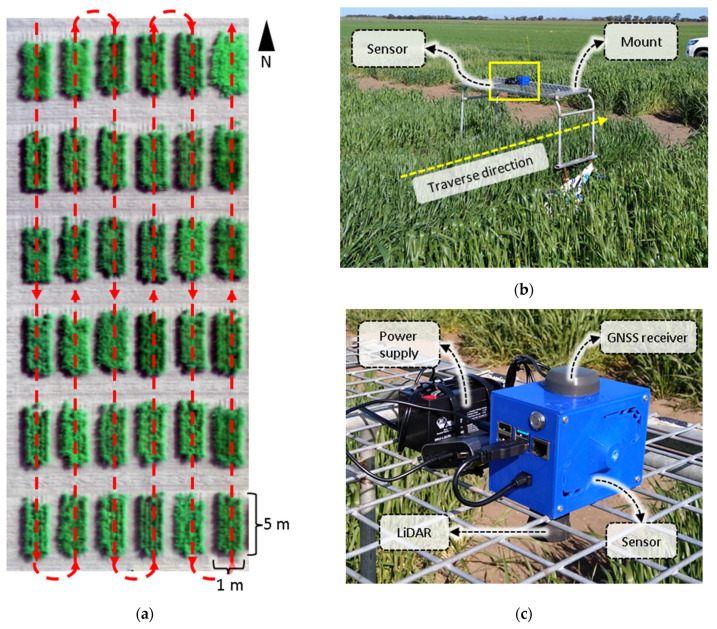
Field experimental and data collection. (**a**) Top view of the field plots, the CropBioMass (CBM) sensor scanned the plots in the trajectory that is specified in red, (**b**) the mobile vehicle mount for the CBM sensor that was used in the experiment and its traverse direction, and (**c**) a close-up view of the CBM sensor hardware highlighting the LiDAR, GNSS receiver, and the power supply units.

**Figure 5 biosensors-12-00016-f005:**
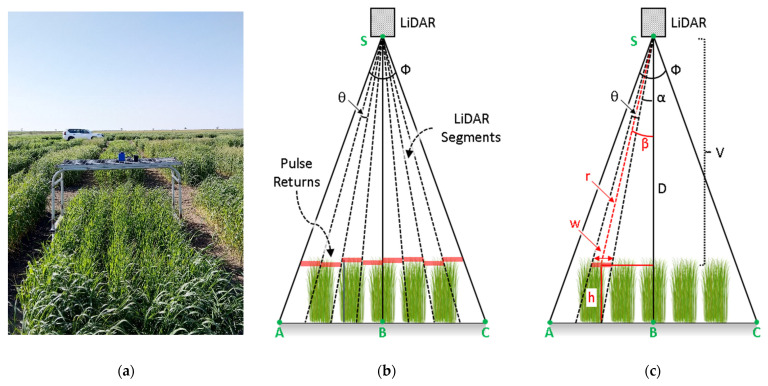
Scanning geometry of the CropBioMass (CBM) sensor with respect to the plots. (**a**) A frontal view of the push-type mobile vehicle with the CBM mounted on it, (**b**) the geometries of each of the eight individual detectors that were stacked inside the CBM sensor with the respective field-of-views (FoVs), and (**c**) trigonometric labeling of the scanning geometry for deriving the height using measure LiDAR ranges.

**Figure 6 biosensors-12-00016-f006:**
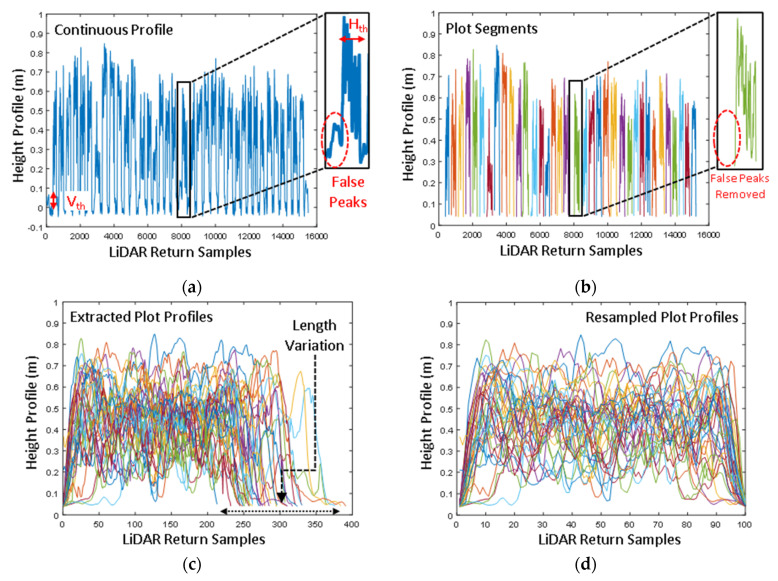
Signal processing to extract the plot profiles. (**a**) The continuous height profile that was measured by the CBM sensor with spurious noise and false peaks, (**b**) removal of the noise, false peaks, and ground profile from the signal; the individual plots were segmented and classified using different colors, (**c**) the extracted plot profiles in high-resolution; the plot profiles varied between 210 and 400 samples due to the variation in the velocity of the mobile platform, and (**d**) the resampled profiles for each of the plot scans.

**Figure 7 biosensors-12-00016-f007:**
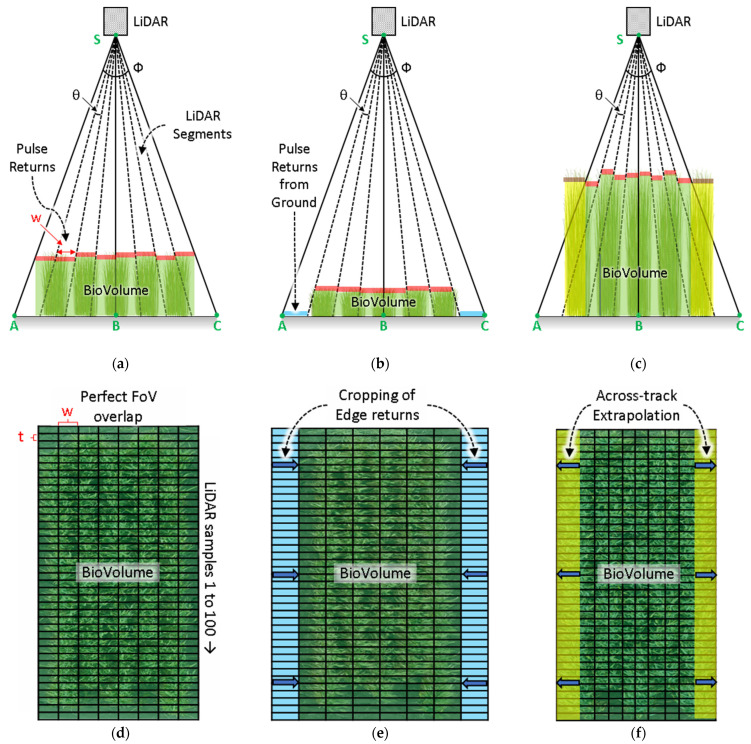
Signal processing to extract BioVolume. The frontal views of the CropBioMass (CBM) sensor geometry for (**a**) medium-, (**b**) short-, and (**c**) tall-sized plots. The top views showing the detector footprint for (**d**), medium-, (**e**) short-, and (**f**) tall-sized plots. The footprint of the individual LiDAR segments vary with the crop height. Short-sized plots are farther from the CBM providing extended FoV coverage area, therefore, the segments on the edge are required to be removed. Tall-sized plots are closer to the CBM providing smaller FoV coverage areas; therefore, extrapolation is required to account for the missed plants on the edge.

**Figure 8 biosensors-12-00016-f008:**
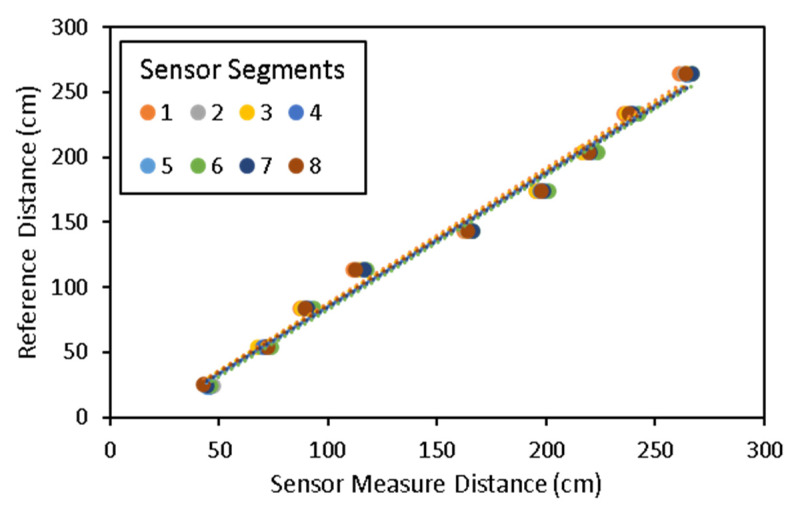
Correlation model between the sensor measured distance and the reference distance. The figure contains the measurement for each of the eight detector elements. Fitted regression models were generated individually for all the detector elements.

**Figure 9 biosensors-12-00016-f009:**
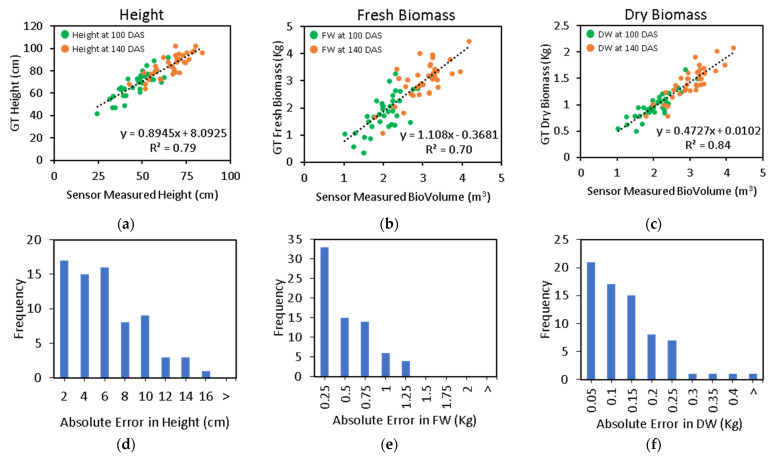
Regression analysis between the CBM sensor readings and the manual measurements (**a**) crop height, (**b**) fresh weight (FW), and (**c**) dry weight (DW). Frequency domain analysis of the absolute residual error for the CBM sensor (**d**) crop height, (**e**) FW, and (**f**) DW.

**Figure 10 biosensors-12-00016-f010:**
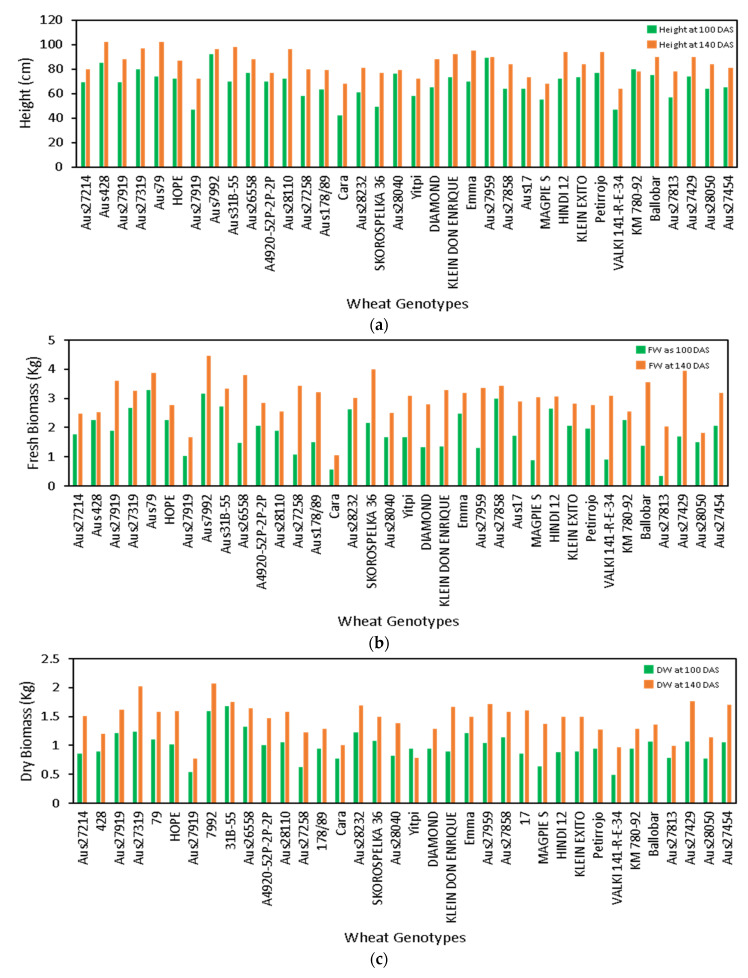
The phenotypic growth profile of the wheat genotypes that were measured between two time points—100 and 140 DAS. (**a**) plant height, (**b**) fresh weight, and (**c**) dry weight. The phenotypic growth profiles that were measured using the CBM sensor aid in the genotypic screening of wheat varieties.

## Data Availability

Not applicable.

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
