# Peer review of "CBM: An IoT Enabled LiDAR Sensor for In-Field Crop Height and Biomass Measurements"

_biosensors, 2021, doi:10.3390/bios12010016_

Round 1
Reviewer 1 Report
Overall, this is a high-quality manuscript that could be accepted for publication in its present form but may benefit from some minor edits and clarification. The introduction gives a concise background on the use of high throughput plant phenotyping biosensors, setting the stage for a new seamless data acquisition and processing IoT system using low data volume LiDAR data for precision plant phenotyping. The methods section contains a thorough description of the LiDAR, Raspberry Pi, and GNSS systems and sound descriptions of all data processing and analysis steps. The results are compelling for the use of this specialized LiDAR system for HTPP in plant phenomic research. I’ve noted below a few places where I have found potential text errors and some minor suggestions the authors may consider, but need not explicitly address in their response.
All line numbers refer to the pdf version of the manuscript (in case the authors are using MS word, which can often feature different line numbers for the same version of a document).
Line 190: Erroneous duplicate use of “triggering the status”?
Line 229: It may be appropriate to specify if the fresh and dry weight included all above-ground biomass?
Line 279: Choose to use either “Although,” or “, but” for this sentence.
Line 288: Here, I’d suggest reiterating that n=8 samples are along the plot width direction for clarity (one coming from each detector). As done on line 314.
Line 298: You may consider discussing, in the discussion section, increasing the standard height of the scanner so that in all cases you have a result where the full FOV captures the edges of the plot width for consistency? Or, how this may need to be altered for use in a practical setting or for other crops, as mentioned on Lines 491-497.
Line 366: The ground truth, as per lines 227-229, was the average of just four random points per plot. It may be worth mentioning that the LiDAR sampling method is arguably more accurate than the actual ground-truthing method, at least for assessing an average crop height within each plot of ~ 750 individual plants (150 per square meter).
Line 369: Here, “actual average canopy height” is used for the ground truth canopy height? It may be better to specify it as such to avoid any confusion, as the LiDAR-derived canopy height is arguably closer to an actual average canopy height.
Line 372: Why gm units for gram, rather than just g or decimal kg? Kg is used later.
Line 453: change “…limiting it seamless adoption…” to “…limiting its seamless adoption”.
FYI: Though not consequential, supplemental tables S1 & S2 were not included in the pdf provided for review.
Author Response
Please find in attachment

Reviewer 2 Report
CBM can operate as an IoT device to enable remote and direct uploading of the captured data on a cloud platform for further processing and analysis. The CBM sensor was deployed on a wheat field trial to evaluate the accuracy and operational efficacy.
Author Response
Please find in attachment

Reviewer 3 Report
The authors report the development of a low-cost integrated sensor system with LiDAR to measure crop biomass and height in the field.
The manuscript has a high significance of content and it is well presented. Considering the merit of the manuscript, I accept it in the presented form.
Author Response
Please find in attachment

Reviewer 4 Report
The manuscript is well written and clearly describes a LiDAR sensor-based system for in-field crop height and biomass measurements. However, the current manuscript requires revisions before it can be accepted for publication.
The main comments are given as follows:
1) From Section 2.1 to Section 2.3, the authors provided the detailed descriptions on system hardware and software architecture, and their design. However, the writeup seems more like a user manual for a product design rather than a scientist journal article because most of the processing steps the authors described are known and should be rephrased (e.g., introduction of the Raspberry Pi 4, LiDAR module, and some basic operations from L168 to L188). The reviewer recommends the authors to rephrase the writeup in these sections and try to limit these three sections within one page.
2) In Section 2.4, the authors provided a laboratory setup for sensor calibration and apply the calibrated parameters to Raspberry Pi to convert the LiDAR imaging unit to real-world distances. However, there are plenty of stuff missing in this section. First of them, why did the authors need this calibration?
Also, if the reviewer understands it correctly, the sensor calibration is an iteration process, where the sensor intrinsic parameters are iteratively optimized based on the reference measurements. In this way, only describing the calibration process from L199 to L202 is not enough. Were any initial parameters the authors feed into the calibration algorithm in this study? How did the authors know when the calibration process ends? Were any specific criteria to control the calibration process such that the output sensor parameters are within the authors’ expectations? What is the meaning of “empirically calibrated” in L208? The reviewer is so confused about the meaning of 9 points * 8 detectors = 72 readings, and the reason why the authors used 72 readings in this study. Please clarify.
3) In L265-L272, the authors provided an approach to remove the effects from ground undulation. Instead of using language to describe the method, the reviewer recommends the authors to explain it using mathematical equations.
4) In experiments, the authors used RMSE and MAE to evaluate the measurement accuracy of the system to manual labels. However, RMSE and MAE are not objective in this case. For example, if the RMSE equals 6.09 cm but the crop height is only 10 cm. In this case, the measurement is not accurate at all because the error rate is achieving 60%. In order to objectively quantify the performance, the reviewer recommends the authors to use the ratios of RMSE and MAE as metrics in the experiments.
Other comments:
1) In L241, L247 and other lines, why did the authors use bold theta? Is the theta a value or a vector? Please clarify.
2) In equation (1), the mid-line dot operator symbol should be used rather than the period symbol.
3) In equation (6), (7) and maybe other areas throughout the paper, the authors repeatedly used cross symbol for number multiplication. In general, the cross symbol is used for matrix multiplication. The reviewer recommends the authors to use mid-line dot for number multiplication.
4) In L321, what is the value for the unit width, Wm,n, and how is this unit width defined?
Author Response
Please find in attachment

Round 2
Reviewer 4 Report
The reviewer thanks the authors to make such revisions. The current manuscript looks better.
As for the main comment 4), "this type of metric is not common in this research" is NOT the reason for the authors to not report it correctly. The limitations of only using the current metric indeed exist in this situation, and the reviewer still suggests the authors to ADD this metric to the experiments. In this way, instead of changing the current metrics to their ratios, both the RMSE, MAE and their corresponding ratios should be provided in the revision.
The other revisions have incorporated the comments from the reviewer, and the reviewer recommends the manuscript for acceptance in Biosensors AFTER adding the RSME and MAE ratios to the experiments.
